# Quantitative capabilities of STXM to measure spatially resolved organic volume fractions of mixed organic/inorganic particles

Matthew Fraund[1], Tim Park[1], Lin Yao[2], Daniel Bonanno[1], Don Q. Pham[1], Ryan C. Moffet[1,†]

[1]Department of Chemistry, University of the Pacific, Stockton, CA, 95211, USA
[2]Department of Chemistry, Beijing Normal University, Beijing, 100875, China
[†]Current address: Sonoma Technology, Petaluma, CA, 94954, USA

*Correspondence to*: Ryan C. Moffet (rmoffet@sonomatech.com)

**Abstract:**

Scanning Transmission X-ray Microscopy coupled with Near-Edge X-ray Absorption and Fine Structure (STXM/NEXAFS) spectroscopy can be used to characterize the morphology and composition of aerosol particles. Here, two inorganic/organic systems are used to validate the calculation of Organic Volume Fraction (OVF) and determine the level of associated error by using carbon K-edge STXM data at 278, 285.4, 288.6, and 320 eV. Using the mixture of sodium chloride and sucrose as one system and ammonium sulfate and sucrose as another, three solutions each were made with 10:1, 1:1, and 1:10 mass ratios (inorganic to organic). The OVF of the organic rich aerosols of both systems deviated from the bulk OVF by less than 1%, while the inorganic rich aerosols deviated by approximately 1%. Aerosols from the equal mass mixture deviated more (about 4%) due to thick inorganic regions exceeding the linear range of Beer's Law. These calculations were performed after checking the data for image alignment, defocusing issues, and particles too thick to be analyzed. The potential for systematic error in the OVF calculation was also tested by assuming the incorrect composition. There is a small (about 0.5%) OVF difference if the organic is erroneously assumed to be adipic acid rather than the known organic, sucrose. A much larger (up to 25%) difference is seen if sodium chloride is assumed instead of ammonium sulfate. These results show that the OVF calculations are fairly insensitive to the choice of organic while being much more sensitive to the choice of inorganic.

**1. Introduction**

Atmospheric aerosols are airborne mixtures of solid and liquid phase components such as soot, inorganic salts, trace metals, and organics (Seinfeld and Pandis, 2006). These aerosols have been shown to cause detrimental health effects upon inhalation and can negatively impact visibility, especially around large cities (Villeneuve et al., 2002). In addition, aerosols currently represent the largest source of uncertainty in radiative forcing from anthropogenic sources according to the 2013 Intergovernmental Panel on Climate Change (IPCC) report (Stocker et al., 2013). Two of the main ways in which aerosols can affect radiative forcing are through aerosol-radiation interactions (also known as the "direct effect") and aerosol-cloud interactions (also known as the "indirect effect"). One of the limitations on the predictive power of global climate models is the dependence that aerosol cloud interactions have on individual particle composition (Pöschl, 2005). Because the complex and varied compositions of aerosols are linked to their impacts on health and the environment, quantitative characterizations of detailed aerosol chemical and physical properties are necessary.

Many methods exist for quantifying the bulk composition of an aerosol sample both in real time ("online") or offline. Offline analysis is most commonly achieved by analysis of filter sample deposits. Characterization with filter samples benefit from a large body of literature detailing standard operating procedures along with a variety of compatible analysis methods (Chow, 1995). Depending on the filter type and composition, a limited elemental analysis can be conducted using, for example, Proton Induced X-ray Emission (PIXE) or Inductively-Coupled Plasma with Atomic Emission Spectrophotometry (ICP-AES) (Artaxo et al., 1993;Menzel et al., 2002). While these methods are highly accurate and precise, they often cannot offer the possibility of quantifying the lighter elements (C, N, O) which often make up the majority of the accumulation mode aerosol.

Online bulk analysis is most often accomplished with relatively complex automated instruments. Thermo-optical analysis is regularly performed and has well established protocols for quantitatively determining fractions of Organic Carbon (OC) and Elemental Carbon (EC) (Karanasiou et al., 2015). The Aerosol Mass Spectrometer (AMS) and the related Aerosol Chemical Speciation Monitor (ACSM) are real-time instruments which can be operated *in situ*, allowing for aerosol events and plume evolution to be studied (Ng et al., 2011). Heavier elements (Al and higher) can be measured in real-time with a portable X-Ray Fluorescence spectrometer with low detection limits ($<40$ ng/m$^3$) (Asano et al., 2017). Laser-Induced Breakdown Spectroscopy (LIBS) is able to detect light and heavy elements down to low ppm-levels in real-time, however shot-to-shot variation hampers this techniques quantitative capabilities (Hahn and Omenetto, 2012;Redoglio et al., 2018;Dudragne et al., 1998). Although many of these quantitative techniques have well defined methods along with, in some sampling locations, long historical records, bulk measurements cannot easily study particle-specific qualities of aerosols.

The challenge of characterizing the composition and source variation within aerosol populations highlights the necessity for quantitative measurement techniques that can determine particle-resolved composition. Many of these techniques are also *in situ* and time-resolved, like the Soot Photometer (SP2) which measures black carbon and any associated coating by way of incandescence and scattering (Raatikainen et al., 2015;Baumgardner et al., 2004;Schwarz et al., 2006). Like its counterpart, the Single Particle Aerosol Mass Spectrometer (SP-AMS) can also provide *in situ* and time-resolved composition but it does so for individual particles (Canagaratna et al., 2007). Single particle laser desorption instruments like the Single Particle Laser Ablation Time-of-flight (SPLAT) mass spectrometer and the Aerosol Time-Of-Flight Mass Spectrometer (ATOFMS) (Spencer and Prather, 2006;Zelenyuk and Imre, 2005;Healy et al., 2013) have been used to obtain particle information about particle composition. Other single particle parameters of interest, like optical properties and particle density, are able to be calculated from ATOFMS data as well (Moffet and Prather, 2005). On-line techniques like these provide useful size and composition information but cannot easily probe the detailed morphology, quantitative single particle composition, or spatially-resolved composition of aerosol particles; for this, microscopic and spectromicroscopic techniques may be better suited. Although microscopy measurements can carry more stipulations (substrate effects (Moffet et al., 2016), sample storage considerations, analysis time, and sometimes large infrastructure requirements), techniques like Scanning Electron Microscopy (SEM), Transmission Electron Microscopy (TEM), Atomic Force Microscopy (AFM), and Scanning Transmission X-ray Microscopy (STXM) can distinguish and classify individual particles or regions therein based on morphology (Ault and Axson, 2016).

In addition to morphology, spectroscopic techniques can be combined with some types of microscopy to study particle composition. For example, high spatial resolution of TEM can be combined with both Energy Dispersive X-ray spectroscopy (EDX, also EDS) and Electron Energy Loss Spectroscopy (EELS) to obtain elemental information about internally mixed particles (Adachi and Buseck, 2008). Nanoscale Secondary Ion Mass Spectrometry (NanoSIMS) has also been applied to aerosol studies, often as a complementary technique to electron microscopy and EDX (Ghosal et al., 2014). This technique can be used to study carbonaceous aerosols (Li et al., 2016) as well as metal-rich aerosols (Li et al., 2017) and is able to provide composition as a function of depth. Another example of a combined spectromicroscopic technique is STXM, which can be

coupled with Near-Edge X-ray Fine Structure spectroscopy (STXM/NEXAFS). This retrieves quantitative elemental composition on a per-particle basis and is well suited for the analysis of C, N, and O. With a spatial resolution of ~30 nm and a spectral resolution of ~150 meV (Kilcoyne et al., 2003;Warwick et al., 2002), STXM can identify the elemental composition of distinct regions within a particle. From this (along with component density) an Organic Volume Fraction (OVF) can be calculated, which can be used to characterize hygroscopicity and has been used in part to quantify the effects of biological activity of laboratory generated sea spray aerosols (Pham et al., 2017). Heavier elements can be difficult to measure in tandem with C, N, and O while using STXM, which has an energy operating range defined by the synchrotron and the design of the STXM. However, heavier elements such as Na and higher can be quantitatively measured using SEM coupled with EDX (Laskin et al., 2006). These two techniques have previously been used in combination on the same set of particles in order to retrieve an elemental composition that is both quantitative and includes lighter and heavier elements (Fraund et al., 2017;Piens et al., 2016).

The presence of an organic component within aerosol populations is important in determining their reactivity and hygroscopic behavior. The amount of organics, and their distribution throughout an aerosol, can affect the reaction rates and equilibrium positions of some heterogeneous reactions.(Worsnop et al., 2002;Maria et al., 2004) Organics can also affect the ability for an aerosol to serve as both cloud condensation nuclei (CCN) or ice nucleating particles (INP) (Cruz and Pandis, 1998;Möhler et al., 2008;Beydoun et al., 2017). Because of the vital role that organics play in affecting aerosol behavior, specifically the effects organics have in changing an aerosol's hygroscopicity, the OVF is used as key piece of data feeding into κ-Köhler theory (Petters and Kreidenweis, 2007). The use of STXM to spatially resolve carbon species in aerosols has been reported since the early 2000s (Russell et al., 2002;Kilcoyne et al., 2003). An automated method for producing spatial maps of these aerosol components was presented in 2010 (Moffet et al., 2010a). The method was further refined in 2016 (Moffet et al., 2016) where the use of carbon maps with only 4 energies was introduced to increase the number of particles analyzed, thereby improving particle population statistics. The quantitative capabilities of this 4 energy mapping method are discussed in the current work by comparing the experimentally determined OVF of two known solutions. Also discussed are the quality control measures necessary to ensure quantitative data, along with the potential for error should they be omitted. Lastly, the uncertainty introduced from assumptions made during OVF calculations was examined. To accomplish this, two systems of inorganic and organic mixtures were studied, each with three formulations of differing inorganic to organic ratios.

## 2. Materials and Methods

### 2.1 Standard Preparation

Standard 100 mL solutions of NaCl/Sucrose and $(NH_4)_2SO_4$/sucrose mixtures were prepared according to Table 1. Standard grade (Sigma-Aldrich, >99% purity) material was weighed using a recently calibrated analytical balance (Torbal, AGN200C) with an accuracy of ± 0.0001g. The powder was quantitatively transferred into a 100 mL volumetric flask (± 0.1

mL) which was then filled with room-temperature Millipore-filtered 18 MΩ distilled water, capped, and inverted to mix. All glassware and utensils were washed and soaked in a nitric acid bath overnight before use.

## 2.2 Sample Production and Collection

After the solutions were prepared, the aerosol generation apparatus in Fig. 1 was assembled. Nitrogen gas at ~ 20 PSI (~140 kPa) was fed into a Collison nebulizer (3 jet MRE, CH Technologies USA) which was filled with one of the standard solutions. The aerosols first passed through an Erlenmeyer flask with a rubber stopper and two stainless steel pieces of tubing in order to collect any large droplets that may have been produced. This flask also had a HEPA filtered air inlet in order to maintain atmospheric pressure and air flow rate. The still humid aerosols next pass through two 66 cm long laboratory-made diffusion driers. The driers consist of a mesh cylinder (2 cm OD) surrounded by a larger Plexiglas cylinder (9 cm OD) with the space between the tubes filled with desiccant (Silica Gel). The dried aerosols are finally directed into a small 4-stage collision impactor (Sioutas Personal Cascade Impactor #225-370, SKC, Fullerton, CA USA) which was loaded with $Si_3N_4$ windows to collect the particles. The four stages had $D_{50}$ size cuts at 2.5, 1.0, 0.5, and 0.25 μm. A small diaphragm vacuum pump was attached to the bottom of the impactor to maintain an air flow of ~9 L/min.

## 2.3 STXM/NEXAFS Data Collection and Analysis

$Si_3N_4$ windows were mounted to an aluminum plate (Kilcoyne et al., 2003) to be imaged at the STXM beamline 5.3.2.2 at the Advanced Light Source (ALS, Berkeley, CA, USA) as well as at the Canadian Light Source (CLS, Saskatoon, SK, Canada). These STXM beamlines have an energy range of 250-780 eV (ALS) and 130-2700 eV (CLS) which allows for the C k-edge to be studied. Soft X-rays were energy selected and then focused to a ~30 nm spot size on the sample surface. A 15x15 μm region containing individual particles was then selected and the sample stage was raster scanned using 40 nm steps. This process was repeated at 4 different energies: A pre and post edge image was taken along with an additional 2 images for C to allow for regions of soot, inorganics, and organics to be determined. The four energies near the carbon edge corresponded to the pre-edge, C=C, COOH, and post-edge regions (278, 285.4, 288.6, and 320 eV respectively).

These groups of images (collectively called a "stack") are first aligned with a method based on Guizar-Sicarios' image registration algorithm (Guizar-Sicairos et al., 2008); this ensures particle positions are constant throughout the stack. Once the stack is aligned, a gamma correction (Reinhard et al., 2010) is applied with each pixel's normalized intensity being raised to an exponent (a γ of 15 is used here) to modify the image contrast in order to detect small, faint particles. Otsu's method is then applied to this enhanced image which automatically differentiates between particles and background (Otsu, 1975). The intensity image is then transformed into an optical density (OD) image on a per-pixel basis using:

$$OD = -\ln\left(\frac{I}{I_o}\right) = \mu\rho t \tag{1}$$

Where *OD* is optical density, *I* is the intensity of the given pixel, and $I_o$ is the background intensity, μ being the mass absorption coefficient, ρ being the density, and t being the thickness of the given pixel.

The additional carbon edge images were used to determine carbon speciation according to previously developed algorithms (Moffet et al., 2010a). With this algorithm, a series of thresholds are used to identify inorganic and organic dominant regions of each particle. These regions are differentiated based on their pre to post edge ratio $OD_{278}/OD_{320}$. Previous work has compared this pre to post edge ratio with the calculated thickness ratios of adipic acid and various inorganics. A pre to post edge ratio of 0.5 was selected as a general thresholding value when the identity of the inorganic isn't known (Moffet et al., 2010a).

Carbon edge images were also used to calculate an organic volume fraction (OVF). This was done by first calculating the thicknesses of both the inorganic and organic components using a previously published method (O'Brien et al., 2015), which is reproduced here. Knowing that the OD at each pixel is due to a mixture of inorganic and organic components, the following equations can be written:

$$OD_{278} = \mu_{278}^I \rho^I t^I + \mu_{278}^O \rho^O t^O \tag{3}$$

$$OD_{320} = \mu_{320}^I \rho^I t^I + \mu_{320}^O \rho^O t^O \tag{4}$$

With $OD_E$ being the optical density at energy E, I and O representing inorganic and organic components respectively. The elemental mass absorption coefficients used here have been retrieved from previously published work.(Henke et al., 1993) By calculating and rearranging $OD_{320} - OD_{278}$ (taking $X^I = \mu_{320}^I/\mu_{278}^I$ for convenience) the thicknesses of the inorganic and organic components can be expressed as:

$$t^O = \frac{OD_{320} - X^I OD_{278}}{(\mu_{320}^O - X^I \mu_{278}^O)\rho^O} \tag{5}$$

$$t^I = \frac{OD_{278} - \mu_{278}^O \rho^O t^O}{\mu_{278}^I \rho^I} \tag{6}$$

Mass absorption coefficients in accordance with published methods (Henke et al., 1993). The densities of sucrose (1.59 g/cm$^3$), NaCl (2.16 g/cm$^3$), and (NH$_4$)$_2$SO$_4$ (1.77 g/cm$^3$) are used in this work. This study takes advantage of the *a priori* knowledge of the inorganic and organic compounds. For the sodium chloride/sucrose system, the mass absorption cross sections for sodium chloride and sucrose are calculated and their known densities are used for calculations. The same thing is done for the ammonium sulfate/sucrose system. Inorganic, organic, and total thickness maps can then be generated, and the OVF for each pixel (or each particle) can be calculated by taking the ratio of organic thickness to total thickness as shown in Fig. 2.

## 2.4 Quality Control of Processed Images

In order to ensure quantitative results from this analysis, each data set was screened for any systematic errors that may have occurred during collection or analysis. The first screening step is to remove stacks with "defocusing issues" where the errors in the zone plate stage positioning result in a sample image that is not in focus. These defocusing issues occurred due to long term wear on the zone plate translation stage. When the sample is not at the focal point of the incoming X-rays, this increases the minimum spatial resolution of the instrument. This also results in unreliable particle morphologies, with many particles taking on a characteristic toroidal or "donut" shape, often seen in unfocused images. Along with major focusing issues, image stacks with a single, slightly defocused image must also be avoided. Field of Views (FOVs) with a single defocused image can present errors in mass determination and C species identification (soot, organic, or inorganic). This is especially apparent near the edges of particles or inorganic cores where defocusing can blur these edges and, for example, misattribute inorganic pixels within an organic coating.

Following this, stacks were reviewed for proper alignment. This was checked by overlaying the aligned images and highlighting pixels which did not match. Any misalignment found was corrected manually before undergoing the automatic particle detection.

The image adjustment done on stacks prior to particle detection can introduce errors. On particularly noisy data a high gamma correction value can accentuate noise peaks causing them to be erroneously labeled as particles, often being only a few pixels wide. Gamma corrections which are too low can also cause the thinner, outer regions of particles to be ignored, instead only detecting the relatively thicker particle cores. Because of this, each FOV was visually inspected for correct particle detection and the gamma correction was adjusted accordingly between $5 \leq \gamma \leq 15$. In addition, a filter was applied after particles are detected which discounts any particles less than 8 contiguous pixels. Detected particles which were cut off by the edge of the image frame were also removed from analysis. An exception was made for particles where only a small portion (less than 8 pixels) appeared to be out of the frame, which were identified manually. These particles were not removed in order to improve particle statistics at the expense of a small error in accuracy.

A final correction was made on any pixels which had an OD >1.5, which is outside of the linear range of Beer's law where Eq. (2) is no longer valid (Moffet et al., 2010b;Wen et al., 2014;Bourdelle et al., 2013). These pixels are from thick/dense regions of the particle, often being from the particle's inorganic center. Because the high OD regions tend to be inorganic cores, and due to the prevalence of cubic NaCl crystals in atmospheric aerosols, the regions are treated by taking the thickness to be equal to the lateral dimension of a cube having the same area as the high OD areas. The number of pixels with an OD >1.5 are added up and the square root of sum is taken, this is then multiplied by the pixel width and the result is used as the particle thickness for all OD >1.5 pixels.

## 3. Results and Discussion

### 3.1 Particle Morphology

Particle mixing state and morphology can potentially impact their effectiveness as either ice nuclei or CCN (Baustian et al., 2013;Baustian et al., 2012;Pöschl et al., 2010;DeMott et al., 2003). Here, particles were produced by nebulization of mixed organic/inorganic solutions of known concentrations. Because each of the solutes are quite soluble in water (with solubilities of 75.4 g/100 mL, 35.9 g/100 mL, and 201.9 g/100 mL for ammonium sulfate, sodium chloride, and sucrose respectively)(Haynes, 2014) and because the nebulization process ensures a well-mixed solution, the resulting droplets are expected to be similar in composition to the homogenous bulk solution and therefore little particle-to-particle variability is anticipated. When qualitatively comparing OVF and carbon speciation maps, most samples exhibited a core-shell morphology that is common in mixed inorganic/organic systems (Shiraiwa et al., 2013;Veghte et al., 2013). A few representative pairs for the sodium chloride/sucrose and ammonium sulfate/sucrose systems are shown in Fig. 3 and Fig. 4 respectively.

### 3.1.1 Sodium Chloride/Sucrose Morphology

For the sodium chloride/sucrose system, across all stages the inorganic rich mixture showed cubic particles with few organic dominant regions according to the C speciation map. In the OVF maps, like the one shown in Fig. 3, a thin coating of organics can be seen surrounding the inorganic centers. This thin coating of organics is not visible in the C speciation map due to the thresholding of the pre to post edge ratio for all pixels exceeding 0.5 as described above. Particles appear to be made up of multiple smaller cubic units which is consistent with scanning electron microscopy images of lab-generated sodium chloride aerosols (Karagulian et al., 2008).

A similar observation can be made for the 1:1 mixture, except with a thicker coating of organics. Of note here is the ability to resolve multiple individual NaCl crystals within some of the aerosols when looking at the OVF maps. Particles collected from the smallest stage (stage D, not shown) can still be seen as an inorganic core with an organic coating in the OVF maps, although it is no longer apparent from the C speciation maps. A few of the particles appear not to exhibit a core-shell morphology but instead look partially engulfed, which is observed for inorganic/organic mixtures under certain conditions (Kwamena et al., 2010). However, this may well be a result of impaction and so it is difficult to comment on how these particles look when airborne.

OVF maps and C speciation maps for the organic rich mixture show circular homogenous particles over all stages. No NaCl dominant inclusions or particles were observed though some inorganic material was detected based on the pre-edge absorption. From the OVF map in Fig. 3, the inorganic phase present is homogenously mixed with the organic phase in this system. Previous studies have shown that high organic concentrations can inhibit the crystallization of inorganic species (Bodsworth et al., 2010;Choi and Chan, 2002).

### 3.1.2 Ammonium Sulfate/Sucrose Morphology

In contrast to the sodium chloride/sucrose system, the inorganic rich mixture for the ammonium sulfate/sucrose system did not show cubic crystals. Instead circular inorganic particles were observed in all stages. Ammonium sulfate particles have been observed with a circular or rounded shape by TEM at these sizes before (Pósfai et al., 1998;Buseck and Posfai, 1999). Pósfai et al suggested that the ammonium sulfate started forming as a polycrystalline solid but then recrystallized. Most of the ammonium sulfate particles observed by Pósfai et al were rounded and, although some particles were aggregates, selected area diffraction (SAED) patterns indicated most were single crystals. In addition, a bumpy irregular surface was documented. Buseck et al showed how ambient ammonium sulfate particles had a coating of organics which filled in these bumps and irregularities. From the OVF maps in Fig. 4 of the current work, a thin coating of organics can be seen in most particles along with a few particles which show a higher than average OVF.

The 1:1 ammonium sulfate/sucrose mixture presented two distinct particle types, seen in both OVF and C speciation maps. A core-shell type of particle is most commonly observed with a defined, rectangular inorganic core surrounded by a thick organic coating. Also seen are circular, fairly homogenous particles which have a pre to post edge ratio > 0.5 according to the C speciation map. The OVF map, however, shows that around 75% of the volume of these particles are attributed to the organic component. This phase separation is discussed in further detail below.

The organic rich mixture of ammonium sulfate and sucrose shows the same behavior as the sodium chloride/sucrose system, with homogenous organic dominant particles. This, as well, is likely attributed to the inhibition of crystallization in concentrated organic solutions. The lack of efflorescence, even at low relative humidity, has been previously observed for ammonium sulfate/organic mixed aerosols with an organic O:C ratio > 0.7 and organic:sulfate mass ratios above 2 (Bertram et al., 2011). In this study, the lack of efflorescence was seen with an organic:sulfate mass ratio of 10:1 and so, with an O:C ratio of 0.91, so this system should be governed by the same principles.

### 3.1.3 Multiple Inorganic Inclusions

For the inorganic rich and 1:1 systems, multiple distinct inorganic inclusions can be seen within individual particles. This may be a result of using the relatively viscous sucrose as the organic component. A similar diffusion dryer setup to the one shown in Fig. 1 has been studied previously reported to dry at a rate of ~99.7% RH/s (Veghte et al., 2013). As a droplet of solution begins to rapidly dry passing through the dryers, its viscosity increases. By becoming increasingly viscous, mass transfer of components within the particle is inhibited (Tong et al., 2011;Bones et al., 2012). Upon reaching a low enough water activity, spontaneous nucleation of the inorganic component begins but diffusion of additional inorganics is hampered by the viscous droplet. As drying continues, more nucleation centers form and crystallize before they are able to combine into a single inorganic core. The formation of single or multiple inorganic inclusions as a result of drying rate has been observed before in less viscous organic/inorganic systems (Fard et al., 2017).

## 3.2 Accuracy of Single-Particle Organic Volume Fractions

Experimental per-particle OVFs for each system and mixture were averaged over all stages and compared with the theoretical OVFs in Fig. 5 and in Table 2. Values for Organic Mass Fractions (OMF) are included as well for completeness sake and for discussion in section 3.3 below. The bulk OVF values were calculated using the composition of the bulk solution from which the particles were generated. For the sodium chloride/sucrose system, the experimental OVFs were underestimated compared to theoretical OVF in all except the inorganic rich mixture which was overestimated. The experimental OVFs for the ammonium sulfate/sucrose mixture are all underestimated as well.

### 3.2.1 1:1 Systems

The largest deviation of the experimental OVF from the OVF calculated from the bulk solution was observed with 1:1 mixtures for both the sodium chloride/sucrose and ammonium sulfate/sucrose systems. Both of these systems were underestimated for similar reasons. Aerosols in both samples contained thick inorganic inclusions surrounded by organics (see Fig. 3 and 4). Many of these thick inorganic crystals were thick enough for their OD to exceed the linear range of Beer-Lambert's law (OD >1.5) and so the correction mentioned above was applied. However, considering that these cores are surrounded with a layer of organics, there is likely a layer above and below which the high-OD correction does not account for. This will lower the apparent volume of organics in those regions and decrease the particle's overall OVF value.

### 3.2.2 Inorganic Rich Systems

The inorganic rich systems for both the sodium chloride/sucrose and the ammonium sulfate/sucrose mixture were slightly overestimated (0.012 and 0.009 respectively, from Table 2). This overestimation may be due to some amount of defocusing, especially in the pre-edge image. Images which were obviously defocused exhibited OVFs much higher (>30% higher for the inorganic systems) than well focused images and were excluded from analysis; however, images with subtler defocusing may still be present. Any defocusing present in the pre-edge images will result in depression of the measured OD, especially around the particle edges. Equation (5) shows that a decreased pre-edge OD will also increase the calculated organic thickness and therefore the OVF as well. As for the potential effects of the high-OD correction, while sodium chloride crystals which exceeded 1.5 OD were present, the organic coatings observed are very thin, making this a minor issue. Instead, if the high-OD correction underestimated the thickness of inorganics present, this could also contribute to the overestimation in OVF for the sodium chloride/sucrose system. Another possible contribution to the slightly high OVF is if any carbonate was incorporated in the standard solutions during nebulization, as this ion will contribute to the carbon post edge value which was assumed to be dependent only on organics. Carbonate picked up from dissolved $CO_2$, however, would only amount to approximately $1 \times 10^{-5}$ g in the 100 mL jar, which would correspond to an erroneous OVF increase of about 0.0005% and so the contribution is negligible (Greenwood and Earnshaw, 2012). The overestimation in the ammonium sulfate/sucrose system was smaller and, unlike with the sodium chloride/sucrose system, was within the margin of statistical error. The inorganic

rich ammonium sulfate system also did not have any issues with thick inorganic regions making it a fairly well-behaved system for STXM analysis. While the decrease in optical thickness of the ammonium sulfate particles compared to the sodium chloride particles could be due to differences in physical height, this is difficult to know given 2-dimensional images. However, the absorption cross section for ammonium sulfate is lower than the cross section for sodium chloride by a factor of 0.35 (Henke et al., 1993), which accounts for most of the difference in optical thickness.

### 3.2.3 Organic Rich Systems

As shown in Table 2, the average OVF value for the organic rich systems are in good agreement with their bulk OVF values, having an error of 0.009 for the sodium chloride/sucrose system and an error of 0.008 for the ammonium sulfate/sucrose system. These errors are the lowest for their respective inorganic/organic systems. Because OVF is calculated using STXM images collected before and after the C absorption edge, it is most sensitive to C containing compounds. In addition, three of the four C edge energies taken were associated with organics. Because of this, organic rich particles may have better defined edges relative to inorganic particles when particle detection is performed. The OVF calculation is thus well suited to organic rich particles like these and because of this, the error in experimental OVF fell within the bounds of statistical uncertainty.

### 3.2.4 Phase Separation in 1:1 Ammonium Sulfate/Sucrose System

For the 1:1 Ammonium Sulfate/Sucrose system seen in Fig. 4, two particle types were observed: particles with a core-shell morphology where the organic regions surround a distinct inorganic core, and homogenous particles where a relatively constant OVF was observed. The presence of both phase-separated and homogenous particles was observed only for the 1:1 Ammonium Sulfate/Sucrose system and was observed across all size ranges. Figure 6 highlights this system as unique compared to the others studied here.

The distribution of OVF values for most systems were Gaussian and centered around the bulk OVF value. The organic rich systems showed little spread due to STXM's sensitivity to carbon. Both inorganic rich systems had wider distributions and the 1:1 systems showed the widest OVF distributions. One issue that can plague particles with crystalline regions is that upon impaction with the substrate the particle can shatter (Mouri and Okada, 1993). Shattering involves small pieces of the particle breaking away, potentially removing organic and inorganic mass from the main particle in difficult to predict ratios. While we do not see small fragments distributed amongst larger ones, small particle fragments are observed in the lowest stage. This may be due to the shattered fragments bouncing upon formation and travelling further down the impactor. The 1:1 Ammonium Sulfate/Sucrose system, however, shows far more spread than any of the others due to the two particle types observed in this system.

This distinction between phase separated particles and homogenous ones has been observed before when mixed ammonium sulfate/polyethylene glycol-400 particles generated from an aqueous solution are quickly dried before collection (Altaf and Freedman, 2017). Some of the particles studied were dried so quickly that a fraction of them were observed to solidify into an amorphous phase rather than nucleate a distinct crystalline phase. Because the diffusion drier setup described

in the experimental section for the current work is drying particles at a similar rate compared to the rate discussed in Altaf et al., 2017, the same two types of particles were observed.

A size-dependent trend was also present in the 1:1 Ammonium Sulfate/Sucrose system, with the homogenous particles tending to be smaller on average than the phase separated ones (see Fig. 7). This behavior was previously observed by Altaf et al in 2016, using an Ammonium Sulfate/Polyethylene glycol mixture. They observed that, depending on the inorganic/organic ratio, the inorganic compound could start to undergo spinodal rather than binodal crystallization. The end result was a size dependence seen in certain inorganic/organic mass ratios, where nucleation of a separate phase became more energetically unfavorable at smaller sizes (Altaf et al., 2016).

The size distributions discussed in Altaf et al., 2016 and Altaf and Freedman, 2017 are on the order of 200 nm, about a factor of 10 smaller than the size distributions observed here. The increase in viscosity from using sucrose as an organic rather than polyethylene glycol may increase the sizes at which phase separated and homogenous particles overlap. This was noted in Altaf et al., 2016, that the components within smaller viscous particles may not have enough time to coalesce into a completely phase separated particle. In addition, rapid drying may also result in the formation of an inorganic shell as the surface of a particle dries without water within the particle able to spread outward fast enough for very viscous particles (Tong et al., 2011).

The presence of two particle types (homogenous and phase separated) do not, alone, account for the spread of OVF seen in Fig. 6 for the 1:1 ammonium sulfate/sucrose system. Although both types of particles are formed from the same bulk solution, and so are assumed to have the same composition, two competing issues in their analysis serve to broaden their OVF distribution. For the phase separated particles, the issue of thick central regions persists. Because the high-OD correction may discount any organic coatings found above or below these regions and so the OVF will be depressed for these particles, this is shown in Fig S1 where all of the high-OD particles are found below the bulk OVF value. Figure S2 shows a related idea where all of the phase separated particles are found below the bulk OVF value because the phase separated particles tend to be the ones with thick inorganic regions. For homogenous particles, having the inorganics distributed throughout the particle rather than concentrated in a core could result in some regions where the inorganics were poorly characterized, thereby raising the OVF. For instance, if any inorganic regions were located near the edge of the particle, the particle detection algorithm could exclude them due to having only 1 STXM image associated with inorganics. As mentioned above, because the organic components have 3 images (some with strongly absorbing transitions), organic regions near the particle's edge are more likely to be better defined. The homogenous particles in Fig. 4 with rough edges and low OVF points to this.

Much of the extreme spread seen in the 1:1 ammonium sulfate/sucrose system, however, is due to the smallest stage (stage D, 0.25 – 0. 5 μm). Figure S3 shows the OVF distribution of this stage and the presence of almost pure inorganic and pure organic particles. This may be the stage where fragments of particles from shattering and particle bouncing are found, with bouncing being a particular issue for viscous particles like these (Virtanen et al., 2010;Dzubay et al., 1976;Jain and Petrucci, 2015;Saukko et al., 2012). Because shattered fragments will not necessarily have the same organic/inorganic ratio as the bulk solution, these particles can have much higher and much lower OVF values. Figure S4 shows an x-ray micrograph

of stage D particles with arrows pointing to potential fragments from shattering. These small particles are irregularly shaped (compared to the numerous surrounding circular particles) and are the particles with the extreme high and low OVF values

## 3.3 Effect of Inorganic and Organic Assumptions on OVF Accuracy

All data shown above has been obtained with the known compounds being used in calculating OVF values. However, studies of ambient samples often lack prior knowledge of the major inorganic and organic species present within individual particles. Previous studies have utilized this OVF calculation for sea spray aerosols, and for these samples sodium chloride and adipic acid were used as proxies. Sodium chloride was chosen as an inorganic due to its prevalence in ocean water and adipic acid was chosen because it has an O:C ratio of 0.66 which corresponds to aged organic aerosol species (Jimenez et al., 2009). Figure 8 and Table 3 shows the result of this assumption for each system studied here.

The sodium chloride/sucrose system shows the effect of changing only the organic assumption from sucrose to adipic acid. Adipic acid has been used before as a proxy for oxidized organic matter based on its O:C ratio (Jimenez et al., 2009). Table 3 shows that the experimental average OVF for each formulation decreased slightly, with the average OVF of the organic rich system decreasing more than the 1:1 system. The inorganic rich system showed very little change in experimental OVF which is expected because the assumed inorganic did not change. The insensitivity of OVF values to the assumed organic has been previously remarked upon using a few other assumed organics as well (Pham et al., 2017).

There is a much more pronounced error introduced by using sodium chloride as a proxy for ammonium sulfate inorganics. Although the sensitivity of OVF to the assumed inorganic is increased compared to the assumed organic, the error is accentuated in this specific case. Because Cl has an absorption edge quite close (~270 eV) to C's absorption edge, including or excluding Cl will result in a significant change in how the pre-edge mass absorption coefficient is calculated and can result in up to a 25% error in OVF. In the case of assuming sodium chloride instead of ammonium sulfate, more of the pre-edge's OD is attributed to the increased absorption coefficient and less to the mass (and therefore the thickness) of the inorganic. This effect inflated the OVF of both the 1:1 and the inorganic rich system. The overestimation of OVF was subdued in the inorganic rich system because OVF is calculated as a ratio between organic and total volume; if the organic volume is small to begin with, the ratio won't be as affected by changes in the total volume.

Although OVFs are of interest due to their utility in κ-Köhler calculations, OMF values are readily obtained and have the benefit of not needing to assume the density of the organic and inorganic components. As far as calculating OMF versus OVF when the composition is known, there is no difference in the error with respect to the bulk solution values (as seen in Table 2). Table 3 also compares the OMF values obtained using the known organic and inorganic composition with the OMF values obtained from the adipic acid/sodium chloride assumption. At first glance, the OMF differences between the known and assumed cases don't share the same trend as seen with OVF values. This is because the error associated with assuming an inorganic and organic composition lies in two places: the calculation of the mass absorption coefficient, and the density. The larger an assumed mass absorption coefficient, the more absorption will be assigned to a specific component. Largely independent from this is the density assumption, which dictates the volume of a specific component. These two values can

serve to affect resulting OVF in the same direction together, or can act separately in opposite directions. Because of this, the effect of removing the assumption of density to calculate OMF instead of OVF changes on a case-by-case basis. For any given set of assumptions, however, the OVF and OMF will always differ by a constant (C) via the following equation

$$C = \frac{(f^I \rho^O + f^O \rho^I)}{\left((f^O + f^I)\rho^I\right)} \tag{7}$$

5 Where $f^x$ and $\rho^x$ represent the mass fraction and density of component x respectively.

The OMF calculations in Table 3 also show that these calculations are more sensitive to the assumptions about the inorganic component than the organic component. The OMF calculations also show that an erroneous mass absorption coefficient assumption will affect the calculation's accuracy even without the assumption of density. In addition, in order to calculate a mass absorption coefficient a molecular formula must be assumed. For organic components, these assumptions are often supported by estimates of O/C or N/C ratios. These constraints, along with the usefulness of the OVF calculation, can make assuming a density worthwhile. The error of using the assumed system of NaCl and adipic acid (with densities of 2.16 and 1.36 g/mL respectively) in the extreme case that the density assumptions are very wrong in opposite directions (say the real composition is $Fe_2O_3$ and pinene with densities of 5.24 and 0.86 g/mL respectively) is approximately a factor of 3.

Note that an appropriate choice for the organic or inorganic proxy for calculation purposes can be guided using peripheral measurements when analyzing ambient samples. Size resolved composition information (either molecular or elemental) can be used to constrain the identity of the components. Additionally, combining another microscopy technique which can probe heavier elements, like Scanning Electron Microscopy with Energy Dispersive X-ray spectroscopy (SEM/EDX), will narrow down the possible inorganics present within individual particles as has been shown previously (Fraund et al., 2017). Also, because the mass absorption coefficient is calculated from the compound's molecular formula, measurements of elemental ratios can serve to improve the OVF value calculation. Because of the erroneous assumption about Cl in the ammonium sulfate/sucrose system discussed above, the change in OVF in this case represents one of the larger errors possible.

4. Conclusions    The OVF values determined experimentally matched the values from the bulk solution well, when the known inorganic and organic compounds are used. Aerosols that are primarily composed of either inorganic or organic seem produce the smallest errors. The OVF of organic aerosols can be determined to within 0.8% under ideal conditions while the OVF of inorganic aerosols can be determined to within about 1%. Additional care must be taken when mixed phase aerosols are present that thick inorganic regions do not compromise the OVF calculation, although OVF can still be calculated to within about 4% even with thick inorganic regions.

The results shown here are most easily attainable after a series of quality control measures have been conducted. Quality control checks for issues including proper alignment, focused images, and accurate particle detection are important. These issues can result in not only a less reliable OVF calculation, but can greatly change the interpretation of an aerosol data set. Most of these issues are best remedied during data collection, though filtering data after the fact can help as well. The

results here also highlight the importance of considering how particle generation and collection factor in to the results. It was observed here that particle shattering and impactor bounce may have contributed to the large spread in OVF values in the 1:1 ammonium sulfate/sucrose system.

The calculation of OVF from OD images necessitates some assumptions which should be examined during data analysis. Regions with high OD (>1.5, outside of the linear range of Beer's law) are again best remedied during data collection by avoiding high OD particles if possible. Although these high OD regions can be approximated, the quantitative nature of this technique can be compromised. This approximation tends to depress the average OVF when organic coatings are present, and so should be kept in mind when interpreting results. To ensure quantitative OVF calculations with tight distributions, which agree with bulk measurements, it is important to focus on mainly carbonaceous particles (to ensure sufficient carbon signal) or particles with thin enough inorganic inclusions (to reduce regions where beer's law is nonlinear). In general, smaller (fine mode) particles will be best suited to this type of calculation. Assumptions about the identity (or at least the molecular formula and density) of the inorganic component can also potentially have a large effect on the calculated OVF. An incorrect assumption can result in an error upwards of 15%. Because most common organic components in aerosols are similar in composition and density, the OVF is much less sensitive to an incorrect assumption here.

Additional spectroscopic images can be used here to great effect. Along with C k-edge data, imaging particles using the nearby Cl, S, Ca, or K edges can help both better define particle boundaries and improve assumptions about the inorganic component. This, however, comes at the cost of particle population statistics as more time is spent on fewer particles. Similarly, the identity of the organic component can be better refined by including more energies while taking C edge data. For example, including an image at 290.1 eV could help remedy the issue mentioned above about carbonate falsely increasing the amount of organics.

With the proper attention paid to the quality of data, STXM can be used to quantitatively determine the OVF of a set of aerosols to within less than 1%. This method of calculating OVF has previously been used on ambient samples as an indirect measure of biological activity in sea water (Pham et al., 2017). When applying this method to ambient samples the analyst should note factors that can affect the accuracy of the results. As an example, volatile organics and inorganics (such as ammonium nitrate) will not be accounted for due to evaporation in the vacuum of the STXM chamber. However, even being predicated on assumptions about the inorganic and organic components, the OVF can be quantitatively determined. Because of this, other STXM results such as the mass fraction of carbon and the absolute mass of carbon (which do not rely on density assumptions) can be determined quantitatively as well. Because STXM offers morphological information along with elemental and molecular composition on a sub-particle basis, it can be a powerful technique for analyzing aerosol populations. If care is taken during data collection and analysis, these quantitative results can be used to develop model parametrizations with some confidence regarding the level of associated error.

**Code Availability:**

MatLab code used for the current work is available as a supplementary .zip file. A set of semi-regularly updated scripts is also available at https://github.com/MFraund/OrganicVolumeFraction_StandardAerosols

**Data Availability:**

The datasets are available upon request to the corresponding author

5 **Author Contributions:**

M.F. led data collection, analysis, and writing of the article and oversaw sample preparation. T.P. did calculations for and prepared the standard solutions and aerosol samples. L.Y. assisted with data analysis and writing. D.B. assisted with sample preparation and collected STXM data. D.Q.P collected STXM data. R.C.M. conceived the experiment, directed standard and sample preparation, collected STXM data, and administered the project. All authors provided input on the project 10 and edited the manuscript.

**Competing Interests:**

The authors declare that they have no conflict of interest.

**Acknowledgements**:

Funding for the data analysis was supported by the U.S. DOE's Atmospheric System Research Program, BER under 15 grant DE-SC0013960. This research used resources of the Advanced Light Source, which is a DOE Office of Science User Facility under contract no. DE-AC02-05CH11231. The authors would also like to acknowledge beamline 5.3.2.2 and its staff: D. Shaprio, D. Kilcoyne, and M. Markus.

Research described in this paper was also performed at the Canadian Light Source, which is supported by the Canada Foundation for Innovation, Natural Sciences and Engineering Research Council of Canada, the University of Saskatchewan, 20 the Government of Saskatchewan, Western Economic Diversification Canada, the National Research Council Canada, and the Canadian Institutes of Health Research. The authors would also like to acknowledge the SM beamline 10ID-1 and its staff: J. Wang, Y. Lu, and J Geilhufe.

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

**Table 1. Calculated masses of each compound needed to make 100 mL of solution. Measured masses in parentheses.**

| System | | Inorganic (g) | Organic (g) |
|---|---|---|---|
| Sodium Chloride / Sucrose | 10:1 | 1.5213 (1.5210) | 0.1521 (0.1523) |
| | 1:1 | 0.7327 (0.7325) | 0.7327 (0.7328) |
| | 1:10 | 0.1185 (0.1185) | 1.1848 (1.1846) |
| Ammonium Sulfate / Sucrose | 10:1 | 1.2742 (1.2748) | 0.1274 (0.1273) |
| | 1:1 | 0.6701 (0.6701) | 0.6701 (0.6701) |
| | 1:10 | 0.1167 (0.1165) | 1.1672 (1.1673) |

**Table 2. Experimental and bulk values for Organic Volume Fraction and Organic Mass Fraction along with their associated absolute error (Relative errors for OVF and OMF are identical to within rounding). Errors with an asterisk cannot be attributed to statistics (95% confidence) alone.**

| System | | Fields of View | Organic Volume Fraction | | | Organic Mass Fraction | | |
|---|---|---|---|---|---|---|---|---|
| | | | Experimental | Bulk | Error | Experimental | Bulk | Error |
| Sodium Chloride / Sucrose | Inorganic Rich | 6 | 0.132 | 0.120 | 0.012* | 0.100 | 0.091 | 0.009* |
| | 1:1 | 3 | 0.538 | 0.576 | 0.039* | 0.467 | 0.500 | 0.033* |
| | Organic Rich | 3 | 0.923 | 0.931 | 0.009 | 0.900 | 0.909 | 0.009 |
| Ammonium Sulfate / Sucrose | Inorganic Rich | 4 | 0.091 | 0.100 | 0.009 | 0.082 | 0.091 | 0.009 |
| | 1:1 | 11 | 0.571 | 0.527 | 0.044* | 0.542 | 0.500 | 0.042* |
| | Organic Rich | 6 | 0.926 | 0.918 | 0.008 | 0.917 | 0.909 | 0.008 |

**Table 3. Experimental and bulk values for Organic Volume Fraction and Organic Mass Fraction under different assumptions about the inorganic and organic component. The difference between OVF values using the known composition versus using sodium chloride and adipic acid is also shown.**

| System | | Organic Volume Fraction | | | Organic Mass Fraction | | |
|---|---|---|---|---|---|---|---|
| | | Assumed Composition | Known Composition | % Difference | Assumed Composition | Known Composition | % Difference |
| Sodium Chloride / Sucrose | Inorganic Rich | 0.132 | 0.132 | 0 | 0.087 | 0.100 | 1.3 |
| | 1:1 | 0.533 | 0.538 | 0.4 | 0.434 | 0.467 | 3.2 |
| | Organic Rich | 0.917 | 0.923 | 0.6 | 0.886 | 0.900 | 1.5 |
| Ammonium Sulfate / Sucrose | Inorganic Rich | 0.198 | 0.091 | 10.8 | 0.132 | 0.082 | 4.9 |
| | 1:1 | 0.723 | 0.571 | 15.2 | 0.589 | 0.542 | 4.7 |
| | Organic Rich | 0.931 | 0.926 | 0.5 | 0.899 | 0.917 | 1.8 |

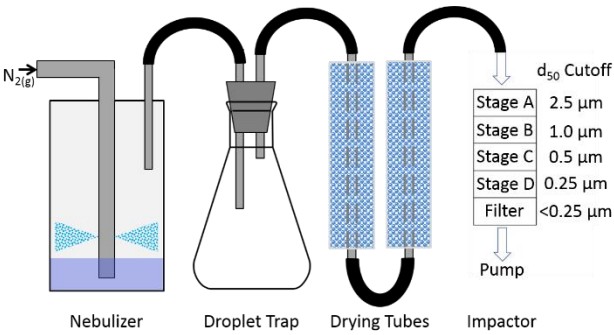

**Figure 1. Schematic of the aerosol generation setup used to nebulize, dry, and collect the lab generated aerosols. Running through the drying tubes are smaller, mesh tubes surrounded by silica gel desiccant (represented in blue).**

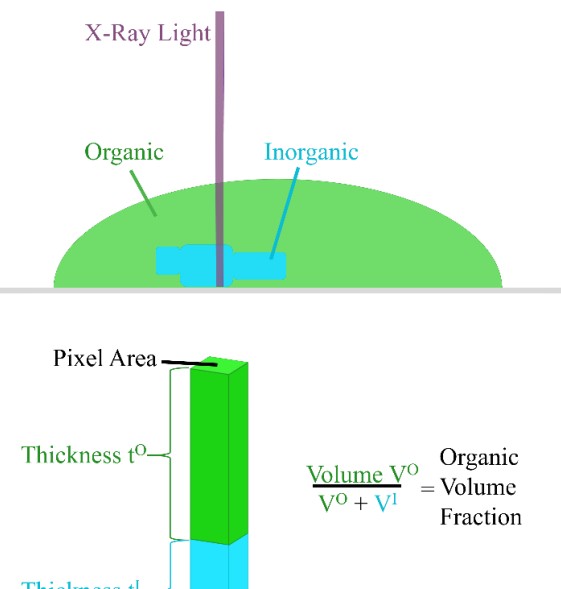

X-Ray Light

Organic     Inorganic

Pixel Area

Thickness $t^O$

$$\frac{\text{Volume } V^O}{V^O + V^I} = \begin{array}{l}\text{Organic}\\ \text{Volume}\\ \text{Fraction}\end{array}$$

Thickness $t^I$

**Figure 2. Visual of Organic Volume Fraction (OVF) calculation with 2 µm particle and 30 nm spot size**

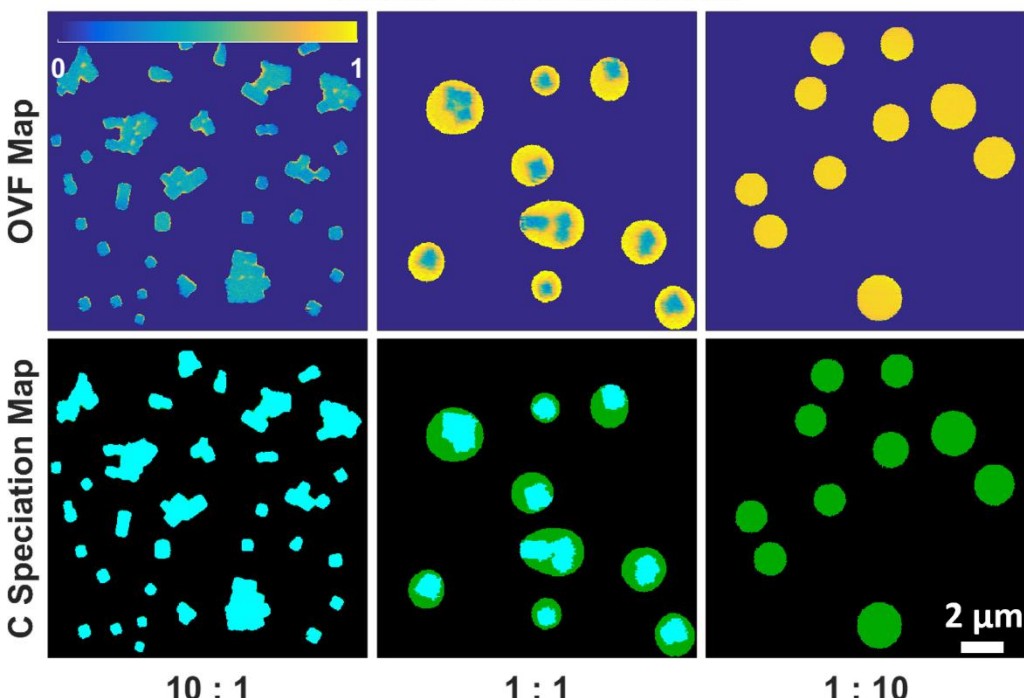

**Figure 3: Representative Organic Volume Fraction (OVF) maps and C speciation maps for the sodium chloride/sucrose system. Samples shown were collected from impactor stage C (0.5–0.25 µm nominal size range). For the C speciation maps, green represents inorganic dominant and blue represents inorganic dominant regions.**

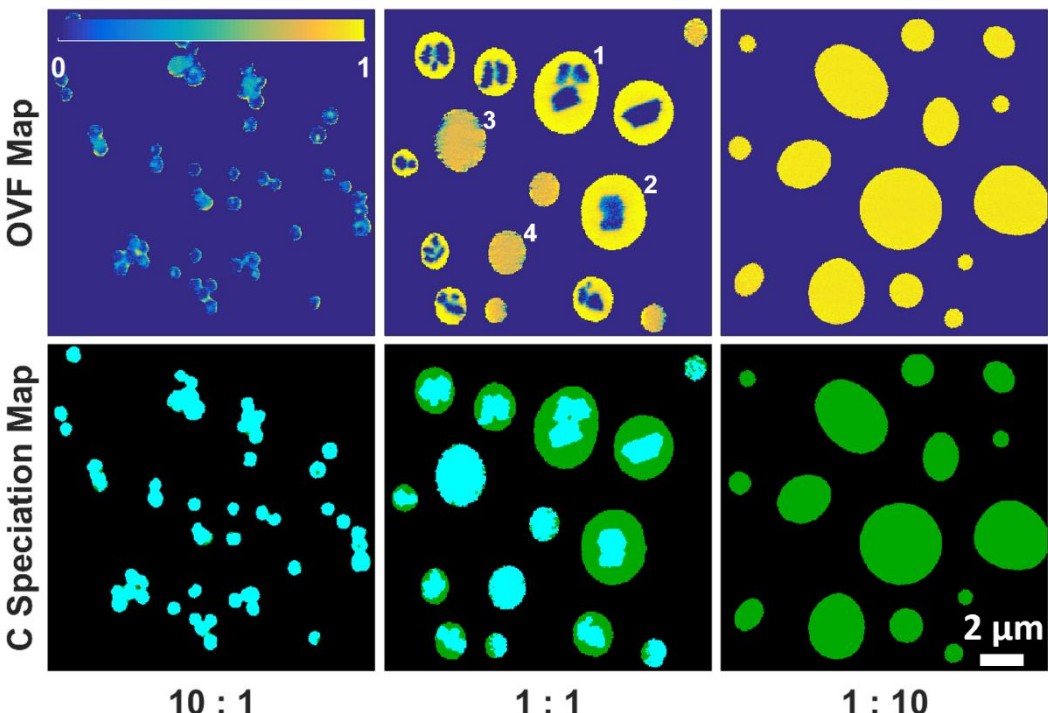

**Figure 4: Representative Organic Volume Fraction (OVF) maps and C speciation maps for the ammonium sulfate/sucrose systems. Samples shown were collected from impactor stage C (0.5–0.25 μm nominal size range). For the C speciation maps, green represents inorganic dominant and blue represents inorganic dominant regions. The 4 particles are labeled in the top middle image, have OVFs of: 0.56, 0.41, 0.77, and 0.79 for particles labeled #1, 2, 3, and 4, respectively.**

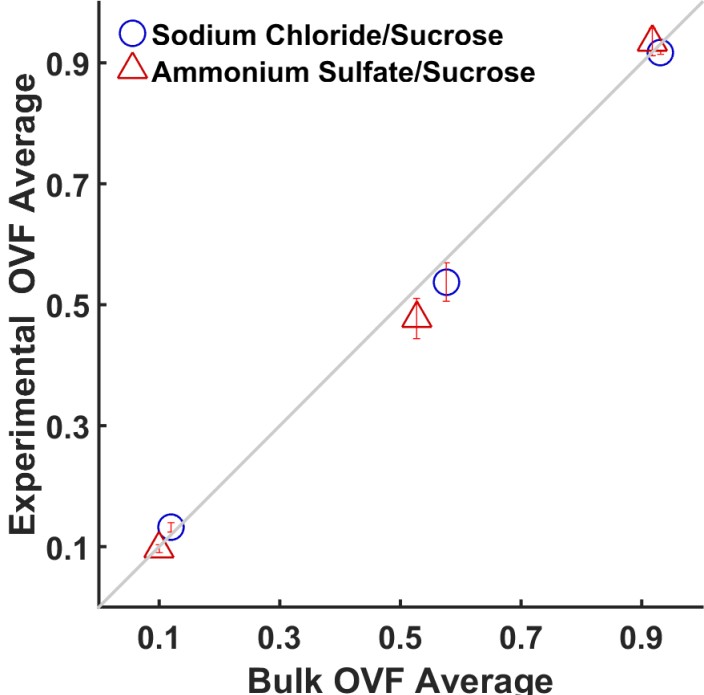

**Figure 5: Correlation between experimentally determined average OVFs and calculated average OVFs for the sodium chloride/sucrose (blue circles) and ammonium sulfate/sucrose (red triangles) systems. A 1:1 line is shown in gray. Error bars represent standard error calculated by StdErr = $(1.96*S)/(N^{1/2})$ where $S$ is the standard deviation, $N$ is the number of particles, and 1.96 is the approximate number of standard deviations encompassing the central 95 % of a student's t-distribution (Skoog et al., 2007). Error in bulk OVF is too small to be shown.**

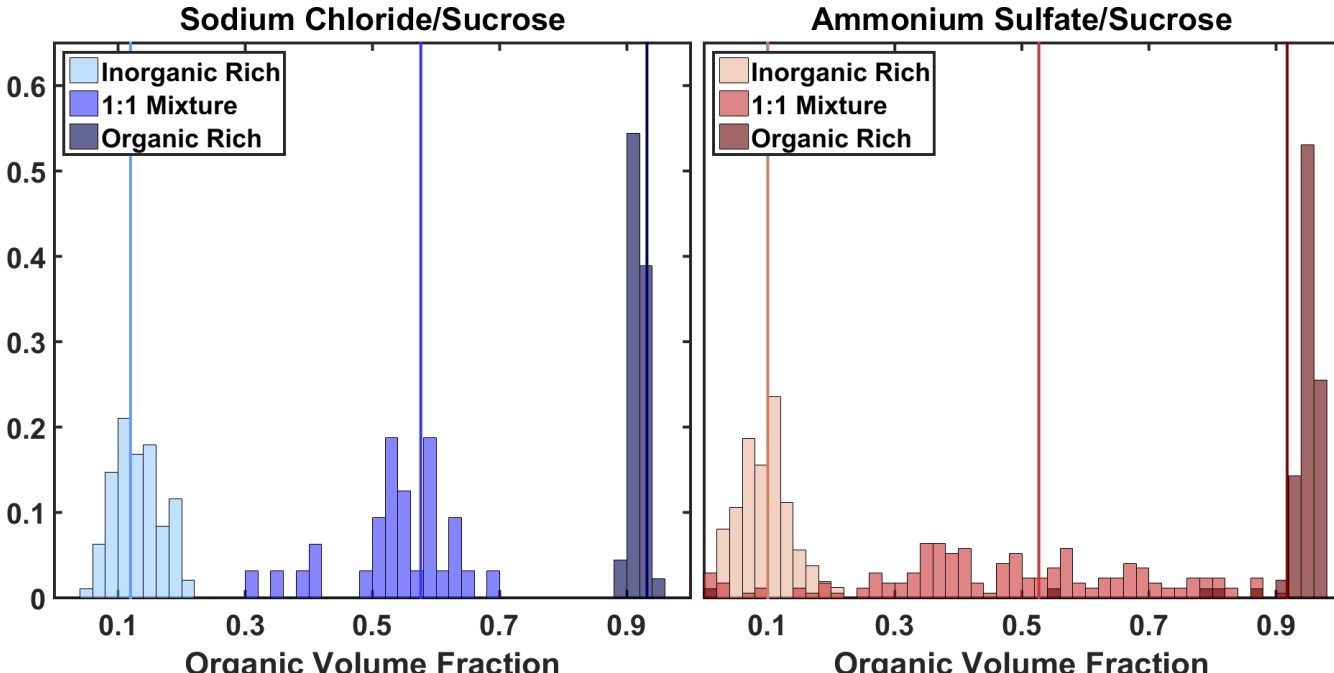

**Figure 6: Organic Volume Fraction (OVF) histograms of both sodium chloride/sucrose (blue) and ammonium sulfate/sucrose (red) systems. The three mass ratios: 10:1 (inorganic rich), 1:1, and 1:10 (organic rich) are shown in different shades of color. The vertical line represents the bulk OVF value.**

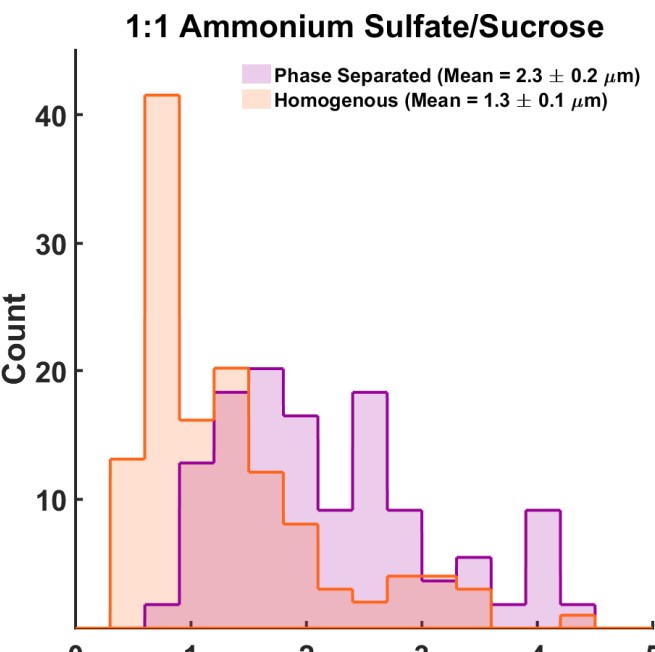

**Figure 7: Circular Equivalent Diameter (CED, also called area equivalent diameter) histogram of homogenous and phase separated particles showing that homogenous particles tended to be smaller than phase separated ones. Particles from stages B, C, and D are included here (2.5 – 0.25 μm)**

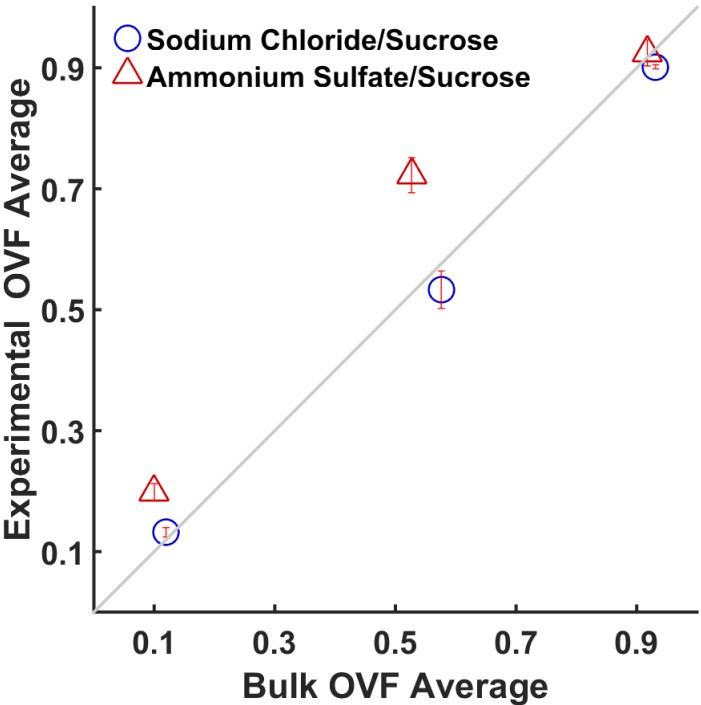

**Figure 8: Correlation between experimentally determined and bulk OVF averages using sodium chloride and adipic acid as the assumed inorganic and organic respectively.**