# Peer review of "Quantitative capabilities of STXM to measure spatially resolved organic volume fractions of mixed organic/inorganic particles"

_Atmospheric Measurement Techniques, 2018_

## Referee Comment (RC1) · Anonymous Referee #2 · 11 Dec 2018

Quantitative capabilities of STXM to measure spatially resolved organic volume fractions of mixed organic/inorganic particles

Fraund et al.,

The study tested STXM to determine organic volume fraction (OVF) in individual particles generated from the lab solutions with 10:1, 1:1, and 1:10 mass ratios (I/O). In recent and near future, STXM is very useful method to study mixing state of individual particles and quantified different light components (containing C, N, O). They studied many different particles and made statistics of STXM data. Finally, they found that the OVF of the organic rich aerosols deviated from the bulk OVF by less than 1%, while the

inorganic rich aerosols deviated by about 1%. These results are helpful to explain the data from STXM in the future. As I noticed that the authors shared their code in Github to transfer the image pixel into organic volume fraction as shown in Figure 2. These code will be useful for the potential user of STXM. I might accept this paper after one minor comments.

L24 compostions, impacts P8L16-20 Posfai et al., (year), Buseck et al., (Year), wrong formation here P7L20 deleted : P12 L3 deleted :

In conclusion section: I would like recommend the authors shorten it. The current one is too long. Seemly, the direct result should be introduced. The first background information should be moved to induction part.

Fig. 1. Drying Tubes. I think that the tube should be one central tube in the dryer instead of all the materials in the dryer. Please check the schematic and make sure it is right. What are blue materials?

Is that possible for authors make Matlab code as the zip. file as supplement associated with the paper in AMT? That would be convenient for the potential user. If the author can provide one simply introduction to process the data, that would be perfect.
* * *

---

## Referee Comment (RC2) · Anonymous Referee #3 · 18 Dec 2018

This review pertains to the manuscript titled "Quantitative capabilities of STXM to measure spatially resolved organic volume fractions of mixed organic/inorganic particles" submitted to the journal, Atmospheric Measuring Techniques. The authors investigated the capability of STXM to quantify the Organic Volume Fraction (OVF) in particles composed by sodium chloride and sucrose and particles composed by ammonium sulfate and sucrose at several inorganic/organic fractions. They use X-ray microscopy and spectroscopy to quantify the OVF of the samples indicated above. They found that STXM can predict fairly well the OVF of aerosol particles when the density is known.

This study is of great importance and it has been done properly and systematically.

This follows the journal guidelines of laboratory measurement techniques for the constituents and properties of atmospheric aerosol particles. The manuscript is timely and fits within the scope of the journal. I recommend the manuscript to be published in Atmospheric Measuring Techniques only after responding to the major and minor comments below.

Major Comments:

1. Why did the authors centered the study in calculating the OVF instead of the Organic Mass Fraction? If the Mass Fraction will be considered, the results are not affected by the density of the organic material. To this reviewer the capability of STXM of calculating the OVF heavily depends on the knowledge of the density of the organic fraction.

2. l. 17-19 page. 9 Why didn't the authors check for contamination of carbonate? Carbonate is easily detectable in the C K-edge region (clear feature at around 290 eV).

Minor Comments:

1. In the abstract it must be indicated that the measurements are done in the K-edge of carbon (maybe even specify the energies used). Otherwise it looks unclear to the reader and it can create confusion.

2. How sure are the authors about the composition of the solution being the exact same composition as the average composition of the particles? (l. 19 and 20 page 7)

3. Why didn't the authors include measurements in any other region of the spectrum apart of the Carbon K-edge?

4. l.30-32 page 10. Why didn't the authors look at chloride or sulfur edge to define better the particle edge in the case of such as problems with that?

5. The characterization of the OVF in atmospheric aerosol particles will be affected by the fact the volatile fraction of organics will evaporate in the STXM chamber in the vacuum. Could the authors comment on that?

Technical comments l. 21 and 22, page 2: Why to specify "previously"?

---

## Referee Comment (RC3) · Anonymous Referee #1 · 20 Dec 2018

I have read "Quantitative capabilities of STXM to measure spatially resolved organic volume fractions of mixed organic/inorganic particles" by Fraund et al. This manuscript discusses how organic volume fractions should be calculated for organic/inorganic particles using analysis of STXM/NEXAFS data. Because this is a powerful single particle technique, the results will be useful to the community. However, I found the paper unsatisfying in two main aspects: 1) Most of their results that they discuss at length in 3.2.1 and 3.2 likely have a lot to do with the viscosity of the system. Yet, viscosity is not discussed. A organic other than sucrose should have been used to minimize the effects of viscosity. 2) There is a huge spread in the organic volume fractions calculated for individual particles, even if the average value is close to the bulk value. The

authors acknowledge this, but do not have a good explanation. It is unclear, then, how useful this technique will be for the calculation of organic volume fraction.

pg 2, line 11-12: It is not clear what "is assumed" means. Did you use sucrose, but then figure out the OVF difference if you assumed sucrose was adipic acid? Or is there a higher error with adipic acid and sodium chloride rather than sucrose and ammonium sulfate?

pg 3: Single-particle mass spectrometers should also be included in this discussion.

I suspect the multiple inorganic rich areas in Figs. 3 & 4 are due to fast drying in a viscous solution. Could you comment further about the origin of having multiple inorganic-rich regions?

pg 7, line 30: NaCl is visible in the C speciation map due to the thin coating of organic compounds on the particles. This could be more clearly explained in the text.

Section 3.2.1: The size regime is much larger than that in Altaf and Freedman 2017 and Altaf et al. 2016. Viscosity likely has a large effect on your results, and should be addressed in the manuscript. See for example Fard, Krieger, and Peter JPCA 2017, 121, 9284. It would have been better to use a much less viscous organic for these studies.

Fig. 6: There is such a spread in the 1:1 ammonium sulfate/sucrose data. Even if the average error is 4% compared to bulk measurements, the determination of OVF is not accurate for this system. The authors don't have an explanation for this large spread. It does not seem like ambient measurements will be very accurate, though perhaps this paper provides insight into how these measurements should be made and what sort of error is associated with them.

[Figure]

---

## Author Comment (AC1) · 13 Feb 2019

The authors would like to thank each of the reviewers for their time and effort spent commenting on our manuscript. The suggestions given by the reviewers were helpful and presented in a collegial fashion. We are happy to present our revised document for the reviewers and hope that we have addressed their concerns. We appreciate their help and insight and think that our work has been improved because of it.

The individual reviewer comments are addressed below, point-by-point, in blue text.

**Comment 1 – Reviewer #2**

L24 compostions, impacts
P7L20 deleted :
P12L3 deleted :
    The above corrections have been made.

P8L16-20 Posfai et al., (year), Buseck et al., (Year), wrong formation here
    Because Buseck and Posfai, 1999 was authored by only 2 people, the in-text citation format that AMT requests is that both authors be listed:
https://www.atmospheric-measurementtechniques.net/for_authors/manuscript_preparation.html

1. In conclusion section: I would like recommend the authors shorten it. The current one is too long. Seemly, the direct result should be introduced. The first background information should be moved to induction part.
    The first paragraph of the conclusion was relocated to the end of the introduction, good suggestion.

2. Fig. 1. Drying Tubes. I think that the tube should be one central tube in the dryer instead of all the materials in the dryer. Please check the schematic and make sure it is right.
    Dashed lines were added to the figure to indicate a central tube surrounded by the blue material. The caption was also updated to explicitly describe the central tube.

2.1 What are blue materials?
    The figure caption was updated to clarify that the blue material is silica gel desiccant.

3. Is that possible for authors make Matlab code as the zip. file as supplement associated with the paper in AMT? That would be convenient for the potential user. If the author can provide one simply introduction to process the data, that would be perfect.
    A .zip file of the current scripts are now available in the supplementary section. I've referenced this in the Code Availability Section P16_L8-9 "Matlab code used for the current work is available as a supplementary .zip file. A set of semi-regularly updated scripts is also available at …"

**Comment 2 – Reviewer #3**

Major Comments:

1. Why did the authors centered the study in calculating the OVF instead of the Organic Mass Fraction? If the Mass Fraction will be considered, the results are not affected by the density of the organic material. To this reviewer the capability of STXM of calculating the OVF heavily depends on the knowledge of the density of the organic fraction.

We agree with this comment. To address the reviewer's concerns, we have calculated mass fractions and added them to Tables 2 and 3. We have also added the following:

P13_L29 – P14_L1-18, two paragraphs were added discussing the Organic Mass Fraction calculations that were added to Tables 2 and 3.

We believe that it is still useful to calculate volume fractions due to their use in common parameterizations. This is stated on P4_L18-20 where we mention the calculation of κ-Köhler theory which uses volume fractions as inputs.

2. L. 17-19 page. 9 Why didn't the authors check for contamination of carbonate? Carbonate is easily detectable in the C K-edge region (clear feature at around 290 eV)..
A sentence was added to P10_L28-30 which includes an estimate for the carbonate mass that would be picked up through $CO_2$ dissolution and it was much less than we were worried about when we mentioned carbonate. Even so, a paragraph was added in the conclusions section about improvements that could be made to the method. P15_L27-28 now has the sentence "For example, including an image at 290.1 eV could help remedy the issue mentioned above about carbonate falsely increasing the amount of organics"

Minor Comments:

1. In the abstract it must be indicated that the measurements are done in the K-edge of carbon (maybe even specify the energies used). Otherwise it looks unclear to the reader and it can create confusion.
The abstract has been updated on P1_L4-5 to clarify this has been done over the carbon K-edge at 278, 285.4, 288.6, and 320 eV.

2. How sure are the authors about the composition of the solution being the exact same composition as the average composition of the particles? (l. 19 and 20 page 7)
We've added a short addition about the solubility of each of the solutes. P8_L5-8 now reads "Because each of the solutes are quite soluble in water (with solubilities of 75.4 g/100 mL, 35.9 g/100 mL, and 201.9 g/100 mL for ammonium sulfate, sodium chloride, and sucrose respectively) and because the nebulization process ensures a well-mixed solution, the resulting droplets are expected to be similar in composition to the homogenous bulk solution"

3. Why didn't the authors include measurements in any other region of the spectrum apart of the Carbon K-edge?
Imaging with STXM is a time consuming process. We often use 4 energies at the carbon K-edge to map the major components of a large number of ambient particles. Here we wanted to validate that method specifically. A sentence was added to the introduction P4_L22-24 which

reads "The method was further refined in 2016 (Moffet et al., 2016) where the use of carbon maps with only 4 energies was introduced to increase the number of particles analyzed, thereby improving particle population statistics"

In the added section in the conclusion about improvements we've also added that additional C-edge information can only benefit the method presented, although it often comes at the cost of collecting data on fewer particles. P15_L25-27 now reads "... [taking extra images], however, comes at the cost of particle population statistics as more time is spent of fewer particles. Similarly, the identity of the organic component can be better refined by including more energies while taking C edge data"

4. l.30-32 page 10. Why didn't the authors look at chloride or sulfur edge to define better the particle edge in the case of such as problems with that?

In the same vein as to why additional carbon images were not taken, we sought to validate specifically the 4 energy mapping method that we've used previously. P4_L24 has a small change where a sentence now reads "The quantitative capabilities of this **4 energy mapping method** are discussed…"

P15_L23-25 now reads "Additional spectroscopic images can be used to great effect here. Along with C k-edge data, imaging particles using the nearby Cl, S, Ca, or K edges can help both better define particle boundaries and improve assumptions about the inorganic component"

5. The characterization of the OVF in atmospheric aerosol particles will be affected by the fact the volatile fraction of organics will evaporate in the STXM chamber in the vacuum. Could the authors comment on that?

In the conclusion section P15_L31-33 we added "When applying this method to ambient samples the analyst should note factors that can affect the accuracy of the results. As an example, volatile organics and inorganics (such as ammonium nitrate) will not be accounted for due to evaporation in the vacuum of the STXM chamber."

Technical comments:
L. 21 and 22, page 2: Why to specify "previously"?

The intro has been updated to read "…also known as…" instead to clear up confusion.

**Comment 3 – Reviewer #1**

1.1 Most of their results that they discuss at length in 3.2.1 and 3.2 likely have a lot to do with the viscosity of the system. Yet, viscosity is not discussed.

Discussion of viscosity was added in a few locations. Section 3.1.3 was added on the effect viscosity may have on particle morphology. Another paragraph in section 3.2.4 also discusses the viscosity and its potential effect on the observed particles.

1.2 organic other than sucrose should have been used to minimize the effects of viscosity.

While we did make solutions using another organic (oxalic acid) with the intent to compare, there were issues with the data we collected and so those samples couldn't be included. There were spine-like inorganic crystals spread among homogenous and phase-separated particles, and the phase-separated particles had inconsistent morphologies. Because we wanted to validate our OVF calculation technique first and foremost, we chose not to include these poorly behaved experimental systems which needed more time and investigation to understand.

2. There is a huge spread in the organic volume fractions calculated for individual particles, even if the average value is close to the bulk value. The authors acknowledge this, but do not have a good explanation. It is unclear, then, how useful this technique will be for the calculation of organic volume fraction.

Statements were added to P11_L23-29 further acknowledging the widened distribution of OVFs for the inorganic rich and 1:1 systems, along with an additional explanation for the widened OVF: "Both inorganic rich systems had wider distributions and the 1:1 systems showed the widest OVF distributions. One issue that can plague particles with crystalline regions is that upon impaction with the substrate the particle can shatter (Mouri and Okada, 1993). Shattering involves small pieces of the particle breaking away, potentially removing organic and inorganic mass from the main particle in difficult to predict ratios. While we do not see small fragments distributed amongst larger ones, small particle fragments are observed in the lowest stage. This may be due to the shattered fragments bouncing upon formation and travelling further down the impactor."

The main answer to this issue was reinforced here in the conclusion section on P15_L12-18 which reads: "Regions with high OD (>1.5, outside of the linear range of Beer's law) are again best remedied during data collection by avoiding high OD particles if possible, though these regions can be approximated, the quantitative nature of this technique can be compromised. This approximation tends to depress the average OVF when organic coatings are present, and so should be kept in mind when interpreting results. To ensure quantitative OVF calculations, which agree with bulk measurements, it is important to focus on mainly carbonaceous particles (to ensure sufficient carbon signal) or particles with thin enough inorganic inclusions (to reduce regions where beer's law is nonlinear). In general, smaller (fine mode) particles will be best suited to this type of calculation."

3. pg 2, line 11-12: It is not clear what "is assumed" means. Did you use sucrose, but then figure out the OVF difference if you assumed sucrose was adipic acid? Or is there a higher error with adipic acid and sodium chloride rather than sucrose and ammonium sulfate?

Clarification was added, P2_L12-13 reads "There is a small (about 0.5%) OVF difference if the organic is erroneously assumed to be adipic acid rather than the known organic, sucrose"

4. pg 3: Single-particle mass spectrometers should also be included in this discussion.

     On P3_L18-22 reference has been made to other single-particle mass spectrometers.  It now reads "Single particle laser desorption instruments like the Single Particle Laser Ablation Time-of-flight (SPLAT) mass spectrometer and the Aerosol Time-Of-Flight Mass Spectrometer(ATOFMS) (Spencer and Prather, 2006;Zelenyuk and Imre, 2005;Healy et al., 2013).  Other single particle parameters of interest, like optical properties and particle density, are able to be calculated from ATOFMS data as well (Moffet and Prather, 2005"

5. I suspect the multiple inorganic rich areas in Figs. 3 & 4 are due to fast drying in a viscous solution. Could you comment further about the origin of having multiple inorganic-rich regions?

     We thank the reviewer for the insightful comment. A new section (3.1.3) was added addressing this issue.  It reads:
"For the inorganic rich and 1:1 systems multiple distinct inorganic inclusions can be seen within individual particles.  This may be a result of using the relatively viscous sucrose as the organic component.  A similar diffusion dryer setup to the one shown in Fig. 1 has been studied previously reported to dry at a rate of ~99.7% RH/s (Veghte et al., 2013).  As a droplet of solution begins to rapidly dry passing through the dryers, its viscosity increases.  By becoming increasingly viscous, mass transfer of components within the particle is inhibited (Tong et al., 2011).  Upon reaching a low enough water activity, spontaneous nucleation of the inorganic component begins but diffusion of additional inorganics is hampered by the viscous droplet.  As drying continues, more nucleation centers form and crystallize before they are able to combine into a single inorganic core. The formation of single or multiple inorganic inclusions as a result of drying rate has been observed before in less viscous organic/inorganic systems (Fard et al., 2017)."

6. pg 7, line 30: NaCl is visible in the C speciation map due to the thin coating of organic compounds on the particles. This could be more clearly explained in the text.

     A sentence of clarification was added to P8_L16-17 which reads: "This thin coating of organics is not visible in the C speciation map due to the thresholding of the pre to post edge ratio for all pixels exceeding 0.5 as described above."

7. Section 3.2.1: The size regime is much larger than that in Altaf and Freedman 2017 and Altaf et al. 2016. Viscosity likely has a large effect on your results, and should be addressed in the manuscript. See for example Fard, Krieger, and Peter JPCA 2017, 121, 9284. It would have been better to use a much less viscous organic for these studies.

     On P12_L11-17 the following paragraph was added:
"The size distributions discussed in Altaf et al., 2016 and Altaf and Freedman, 2017 are on the order of 200 nm, about a factor of 10 smaller than the size distributions observed here.  The increase in viscosity from using sucrose as an organic rather than polyethylene glycol may increase the sizes at which phase separated and homogenous particles overlap.  This was noted in Altaf et al., 2016, that the components within smaller viscous particles may not have enough time to coalesce into a completely phase separated particle.  In addition, rapid drying may also result in the formation of an inorganic shell as the surface of a particle dries without water within the particle able to spread outward fast enough for very viscous particles (Tong et al., 2011)."

8. Fig. 6: There is such a spread in the 1:1 ammonium sulfate/sucrose data. Even if the average error is 4% compared to bulk measurements, the determination of OVF is not accurate for this system. The authors don't have an explanation for this large spread. It does not seem like ambient measurements will be very accurate, though perhaps this paper provides insight into how these measurements should be made and what sort of error is associated with them.

The paragraph discussing the spread of the 1:1 ammonium sulfate/sucrose system starting on P11_L17 mentions thick inorganic regions (often seen in phase separated particles) with high OD (exceeding the linear range of beer's law). To account for this, we approximate these regions as a cubic crystal of inorganics but in doing so discount the small layer of organics above and below these regions.

We've also added a supplementary section with some figures taking a closer look at this system and its spread. These figures support our explanation that high-OD particles (often the phase separated ones) contribute to lowering the OVF. We've also observed that our smallest stage particles are causing the extreme spread (OVF's near 0 or near 1) and we think this is a consequence of fragments of larger particles shattering and bouncing, making their way down the impactor.

In addition, to the conclusion section was added a statement about the proper circumstances in which quantitative OVF is to be expected. P15_L15-18 says: "
[revised manuscript text omitted]

[Figure]

**Figure S1. Histogram of particles from stages B and C from the 1:1 ammonium sulfate/sucrose system. The OVF of particles with regions exceeding an Optical Density (OD) of 1.5 are shown in red. The black vertical line is the OVF expected from the bulk solution.**

[Figure]

**Figure S2. Overlapping histograms of both phase separated and homogenous particles seen in stages B and C for the 1:1 ammonium sulfate/sucrose system. The black vertical line is the OVF expected from the bulk solution.**

[Figure]

**Figure S3. Histogram of OVFs for only stage D particles for the 1:1 ammonium sulfate/sucrose system showing that the extreme spread originates mainly from these particles. The black vertical line is the OVF expected from the bulk solution.**

[Figure]

**Figure S4. Average Optical Density (OD) micrograph of 4 carbon k-edge energies (278, 285.4, 288.6, and 320 eV) for an example field of view from stage D of the 1:1 ammonium sulfate/sucrose system. The average OD image provides good contrast for visualizing inorganic and organic particles. Blue arrows point to some irregularly shaped particles which suggests fragments from particle shattering.**